# DATA-EFFICIENT OPERATOR LEARNING VIA UNSUPERVISED PRETRAINING AND IN-CONTEXT LEARNING

**Wuyang Chen**[*,1,2]**, Jialin Song**[*,2]**, Pu Ren**[3]**, Shashank Subramanian**[3]**,
Dmitriy Morozov**[3]**, Michael W. Mahoney**[1,2,3]

[1] Department of Statistics, University of California, Berkeley
[2] International Computer Science Institute
[3] Lawrence Berkeley National Laboratory

## ABSTRACT

Recent years have witnessed the promise of coupling machine learning methods and physical domain-specific insight for solving scientific problems based on partial differential equations (PDEs). However, being data-intensive, these methods still require a large amount of PDE data. This reintroduces the need for expensive numerical PDE solutions, partially undermining the original goal of avoiding these expensive simulations. In this work, seeking data efficiency, we design unsupervised pretraining and in-context learning methods for PDE operator learning. To reduce the need for training data with simulated solutions, we pretrain neural operators on unlabeled PDE data using reconstruction-based proxy tasks. To improve out-of-distribution (OOD) generalization, we further assist neural operators in flexibly leveraging in-context OOD examples, without incurring extra training costs or designs. Extensive empirical evaluations on a diverse set of PDE equations demonstrate that our method is highly data-efficient, more generalizable, and even outperforms conventional vision-pretrained models.

## 1 INTRODUCTION

Recent advancements in machine learning methodology have shown promise in solving partial differential equations (PDEs) (Han et al., 2018; Raissi et al., 2019; Kovachki et al., 2023; Li et al., 2021a; 2020; Lu et al., 2019; Krishnapriyan et al., 2023; Hansen et al., 2024). A significant development is neural operator learning, which can effectively capture complex, multi-scale dynamic processes in relatively simple settings (Li et al., 2021a;b; Takamoto et al., 2022; Subramanian et al., 2023). However, neural operator methods tend to suffer from a problem common to other neural network methods, namely the **need for enormous quantities of data**. High-fidelity numerical simulations are computationally costly or even infeasible for many applications (Sirignano et al., 2020). For example, an extreme-scale simulation of magnitude 7.0 earthquake at frequencies up to 10 Hz in the San Francisco Bay Area requires 3,600 Summit GPU nodes and 42.7 hour run time (McCallen et al., 2021).

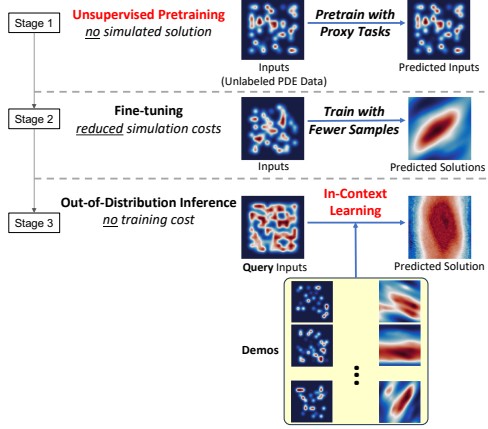

Figure 1: Overview of our data-efficient neural operator learning (contributions highlighted). Stage 1: Unsupervised pretraining only on unlabeled PDE data. Stage 2: Fine-tuning with reduced simulation costs of PDE data. Stage 3: Test-time in-context learning to further improve the neural operator's out-of-distribution generalization with no training costs.

Motivated by these data-efficiency challenges, recent work in machine learning, in particular natural language processing (NLP) and computer vision (CV), has focused on two general strategies for

---

[*]Equal contribution.

reducing the cost of collecting or generating labeled data for training: **unsupervised (self-supervised) pretraining** (Chen et al., 2020; Brown et al., 2020; Devlin et al., 2018) and **in-context (few-shot) learning** (Brown et al., 2020; Logan IV et al., 2021).

However, unsupervised pretraining is still largely under-explored in Scientific Machine Learning (SciML). Although ICL has been introduced to operator learning Yang et al. (2023a;b); Liu et al. (2023), it introduces extra training costs and an explicit number of few-shot examples, deviates from the successful ICL practice in NLP, where language models are trained with simple token-prediction tasks and can easily scale up to *any* number of few-shot examples during inference. Therefore, our core questions are:

*Q1: How to design unsupervised pretraining for operator learning to reduce data simulation costs?*
*Q2: How to design a simple ICL for neural operators to achieve data-efficient OOD generalization?*

To achieve data efficiency in SciML, we propose unsupervised pretraining and ICL for neural operator learning (Figure 1). We first demonstrate that with unsupervised pretraining, we not only improve neural operators trained with more simulated data, but also outperform off-the-shelf pretrained checkpoints from other popular domains. Then, to further improve the data efficiency during OOD inference, we design a simple ICL strategy that is flexible and can scale up to any number of unseen supporting examples ("demos"). This approach also introduces zero overhead during training: one just keeps the standard training pipeline, and our ICL can be seamlessly plugged in (without further fine-tuning) for data-efficient OOD inference. In more detail, we summarize our main contributions:

1. We introduce unsupervised pretraining for data-efficient neural operator learning. We show that unsupervised pretraining on unlabeled PDE data can achieve better performance than models trained with more simulated data and publicly-available checkpoints pretrained on other benchmarks (such as CV), demonstrating the importance of pretraining on PDE domain data.

2. We propose a flexible ICL approach that can improve the OOD generalization of neural operators, without introducing any extra costs during training.

3. We provide detailed experiments on diverse PDEs, demonstrating that we can achieve significant savings in PDE simulations and improve the performance of neural operators in OOD contexts.

## 2 METHODS

In our framework (Figure 1), we propose to first pretrain the model with unsupervised pretraining (Sec. 2.1), which will contribute to the data efficiency and reduced PDE simulation costs. In OOD scenarios during inference, we test our models with ICL to avoid further fine-tuning costs (Sec. 2.2).

### 2.1 UNSUPERVISED PRETRAINING

The core idea of unsupervised (or self-supervised) pretraining is to train a neural network with properly designed proxy tasks, which do not require labeled data but are highly related to objectives of supervised learning. Unsupervised pretraining on unlabeled data has never been explored in PDE operator learning in SciML, mainly due to the following two unresolved questions: 1) What kinds of unlabeled data can we use to train neural operators? 2) How can we design proxy tasks for PDE data?

#### 2.1.1 UNLABELED PDE DATA

Let us consider the second-order linear differential equation as a general example. It is formulated as

$$\sum_{i,j=1}^{n} a_{ij}(x)u_{x_i x_j} + \sum_{i=1}^{n} b_i(x)u_{x_i} + c(x)u = f(x).$$

$x \in \mathbb{R}^n$ represents physical spaces that varies in different systems ($n = 3$ for 2D time-dependent PDEs), and the coefficients (or physical parameters) $a_{ij}, b_i, c$ are known from the physical process; $u$ is the target solution and $f$ denotes an external forcing function Nandakumaran & Datti (2020).

- For time-independent equations, following previous works (Li et al., 2021a; Subramanian et al., 2023), our unlabeled PDE data include physical parameters ($a_{ij}, b_i, c$), forcing functions ($f$), and coordinates (grids of the discrete version of the physical space).

- For time-dependent equations (e.g. forecasting problems Takamoto et al. (2022); McCabe et al. (2023)), our unlabeled PDE data include initial snapshot $u_0(x)$ without simulating the temporal dynamics, and contain sufficient information for defining PDE systems.

There are two main reasons for pretraining only on unlabeled PDE data, as discussed below.

**Cheap Generation of Unlabeled PDE Data.** Only generating unlabeled PDE data and snapshots without temporal dynamics will be much cheaper than simulating solutions, making our unsupervised pretraining highly feasible in practice. This pretraining strategy will be very data-efficient and can avoid the high computational cost of simulating complex time-dependent equations for massive high-fidelity labeled solutions (Table 3 in Appendix E).

**Benefits of Pretraining on Unlabeled PDE Data.** Beyond the cheap generation, pretraining on unlabeled PDE data has the following benefits. First, *regularization against overfitting in the low-data regime*. When training on extremely low volumes of data, neural operators tend to overfit with poor generalization. Instead, unsupervised pretraining can strongly regularize the model towards better generalization, without the need for any early-stopping heuristics. Second, *more meaningful representations*. Pretraining on unlabeled PDE data can help models extract useful representations for subsequent operator learning. We defer our results and experimental details in Figure 5 in Sec. G.

### 2.1.2 PROXY TASKS

We input unlabeled PDE data to our neural operators, and after a decoder network we force the output to be close to the input. We choose two variants of reconstruction as our core proxy tasks.

**Masked Autoencoder.** Masked autoencoders (MAEs) have been shown to be scalable self-supervised learners (He et al., 2021). The method is conceptually simple: remove a portion of the input data, and learn to predict the removed content. The motivation for using MAEs is that PDE dynamics are invariant to sparse sensing of the full field scientific data (Brunton et al., 2020; Erichson et al., 2020; Brunton et al., 2020). The invariance to sparse sensing of scientific data will facilitate the robustness of the representations from MAE.

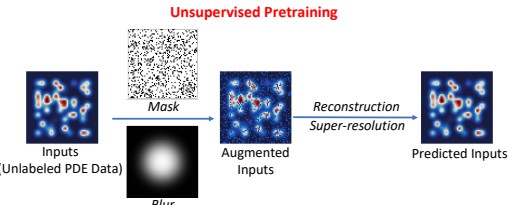

Figure 2: Overview of unsupervised pretraining via MAE and super-resolution. During pre-training, in the input unlabeled PDE data, a random subset of spatial locations of (e.g., 70%) is masked out, followed by a Gaussian blur. After the encoder and decoder, the full set of input is required to be reconstructed.

Therefore, we consider MAE as a proxy task to compute the mean squared error (MSE) between the reconstructed input and its partial observation. We compute the loss only on masked areas; if no mask is applied (i.e., a vanilla autoencoder), the MSE loss will be applied to all spatial areas.

**Super-resolution.** Super-resolution (SR) techniques have emerged as powerful tools for enhancing data resolution, improving the overall quality and fidelity of data and retrieving fine-scale structures. It is also a popular task on PDE learning (Chung et al., 2023; Ren et al., 2023). The motivation for using SR is that numerical solutions of PDEs are expected to exhibit invariance to filtering blur or different resolutions of initial conditions (Kochkov et al., 2021; Vinuesa & Brunton, 2022).

We introduce SR as another proxy task. We apply a Gaussian filter to blur the input, and the autoencoder will reconstruct the high-resolution unlabeled PDE data. Instead of applying a fixed blurring, we randomly sample the variance of the Gaussian filter from a certain range as augmentations.

After pretraining, we fine-tune on simulated PDE solutions (Appendix B): two time-independent PDEs (Poisson, Helmholtz) and two time-dependent PDEs (Reaction-Diffusion, Navier-Stokes).

### 2.2 IN-CONTEXT LEARNING

Out-of-distribution (OOD) generalization is critical, not only to SciML but also across problems in AI for science (Zhang et al., 2023). We propose a simple in-context learning (ICL) of neural operators during inference (Algorithm 1): given an unseen query input, the model is also provided with a few supporting examples (dubbed "demos"), together with their ground-truth solutions.

**1. Similarity by Prediction.** We find spatial-wise and temporal-wise similar demos by calculating their distance in the output space. That means, for two input locations over the spatial and temporal

resolution, if we find their outputs of the trained neural operator similar, then we treat them as similar samples. We assume demos share the same distribution of physical parameters with the query.

**2. Aggregation.** For each spatial-temporal location of the query, after finding its similar samples in demos, we aggregate and average their solutions as the prediction.

# 3 EMPIRICAL RESULTS

## 3.1 UNSUPERVISED PRETRAINING ENABLES DATA-EFFICIENT OPERATOR LEARNING

We first demonstrate that, by leveraging unsupervised pretraining, neural operators can achieve improved relative errors with less simulated data (Figures 3). Specifically, our method can save $5 \times 10^2 \sim 8 \times 10^5$ simulated solutions across diverse PDE systems.

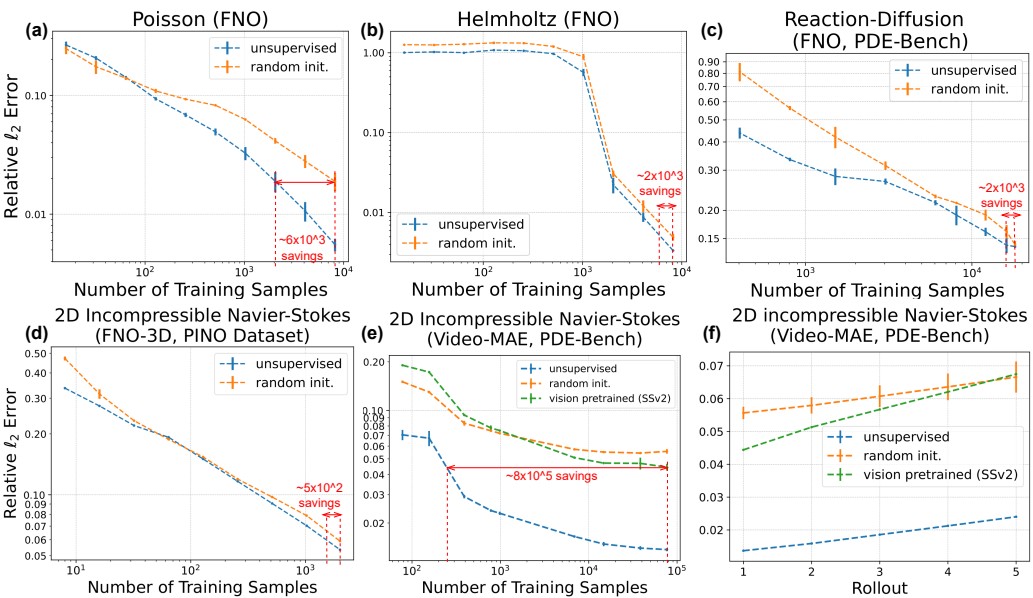

Figure 3: Pretraining neural operators on unlabeled PDE data improves its performance and data efficiency on Poisson (**a**), Helmholtz (**b**), Reaction-Diffusion (**c**), and Navier-Stokes (**d** and **e**, with relative errors at different unrolled steps shown on **f**). "random init.": models are trained from scratch with random initialization. "vision pretrained (SSv2)": fine-tuning from the publicly available checkpoint for Video-MAE (pretrained on computer vision dataset SSV2 (Goyal et al., 2017) for video understanding). Savings of the number of simulated PDE data (when "random init." achieves the best test error) are shown in red.

## 3.2 IN-CONTEXT LEARNING (ICL) ENABLES DATA-EFFICIENT OOD GENERALIZATION

In OOD settings, models will be tested on PDE data simulated with unseen physical parameters. As shown in Figure 4, when we flexibly scale up the number of demos, we can keep improving FNO's OOD generalization on diverse PDEs. Notably, we introduce zero training overhead: we keep the standard training pipeline, and our ICL can be seamlessly plugged in during OOD inference.

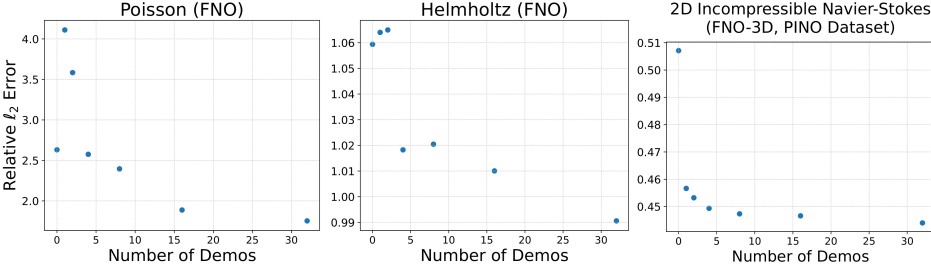

Figure 4: ICL for FNO in OOD testing. Relative errors decrease as we increase the number of in-context demos.

## 4 CONCLUSION

We focus on enhancing the data efficiency of solving neural operators for PDEs. Our key contributions include the introduction of unsupervised pretraining for neural operator learning and a flexible ICL approach that enhances OOD generalization. Through extensive experiments, we demonstrate that our method is not only data-efficient but also more generalizable. We anticipate that our work would inspire the SciML community to explore more on solving the heavy simulation costs and unsatisfactory OOD generalization of neural operators.

## ACKNOWLEDGEMENTS

This work was supported by the U.S. Department of Energy, Office of Science, Office of Advanced Scientific Computing Research, Scientific Discovery through Advanced Computing (SciDAC) program, under Contract Number DE-AC02-05CH11231 at Lawrence Berkeley National Laboratory. This research used resources of the National Energy Research Scientific Computing Center (NERSC), a U.S. Department of Energy Office of Science User Facility located at Lawrence Berkeley National Laboratory, operated under Contract Number DE-AC02-05CH11231 using NERSC award ASCR-ERCAP0023337.

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

# A    RELATED WORKS

## A.1    MACHINE LEARNING FOR SCIENTIFIC MODELING

There has been a long history of using machine learning-based methods to model (often nonlinear) physical scientific phenomena  (Lagaris et al., 1998; 2000; Chen & Chen, 1995b;a).

A representative line of work is so-called physics-informed neural networks (PINNs) (Raissi et al., 2019; Zhu et al., 2019; Geneva & Zabaras, 2020; Gao et al., 2021; Ren et al., 2022), which try to incorporate physics in neural networks by including the differential form of the PDE as an additional physics loss regularization term.  However, this paradigm is confined to specific PDE scenarios (e.g., fixed PDE coefficients) instead of being more physics-agnostic. Moreover, recent work has highlighted fundamental issues with PINN-based methods (Krishnapriyan et al., 2021; Edwards, 2022). On the other hand, operator learning methods, including Fourier Neural Operators (Li et al., 2021a; 2020; Kovachki et al., 2023) and Deep Operator Network (Lu et al., 2019), have achieved progress in approximating the solution operators of PDEs. Although these data-driven approaches show promise in learning PDE solutions, they (like many neural network-based methods) rely on vast quantities of high-fidelity labeled data. For operator learning, such data are usually computationally expensive to simulate (Raissi et al., 2019; Brandstetter et al., 2022; Zhang et al., 2023).

## A.2    UNSUPERVISED PRETRAINING AND FOUNDATION MODELS

Unsupervised (or self-supervised) pretraining is a key method in CV and NLP to achieve meaningful representations (Chen et al., 2020), data-efficient fine-tuning (Houlsby et al., 2019), and foundation models (Bommasani et al., 2021). In CV, contrastive learning learns meaningful features by distinguishing between similar (positive) and different (negative) samples (Oord et al., 2018; Tian et al., 2020; Chen et al., 2020; Misra & Maaten, 2020; He et al., 2020). Masked Autoencoder (MAE) (He et al., 2021) utilizes a reconstructive approach where parts of the input are masked and the model learns to predict masked parts. In NLP, among the most prominent works are large language models (LLMs) such as GPT (Brown et al., 2020; Radford et al., 2018; 2019) and BERT (Devlin et al., 2018), which leverage next token prediction for pretraining. The remarkable abilities (Wei et al., 2022) exhibited by foundation LLMs underscore the critical role of scale in achieving advanced capabilities that are not evident at smaller scales. Similar directions also show progress in SciML.  Brandstetter et al. (2022) and Mialon et al. (2023) propose to create augmented views in the solution space via Lie Symmetries. Most notably, Subramanian et al. (2023) study the scaling behavior of supervised pretraining and OOD generalization, charting directions for foundational models for SciML. Lanusse et al. (2023) targets learning astronomical foundation models with cross-modal contrastive learning. McCabe et al. (2023) builds large task-agnostic models with a broad understanding of common physical behavior to serve as foundation models for SciML.[1]

## A.3    IN-CONTEXT LEARNING

In-context learning is a promising paradigm in deep learning that helps deep networks generalize to unseen domains with a few in-context examples. Early works in computer vision seek to learn feature-level correspondence between the target and a few "shots", such that models can generalize to open-set unseen objects (Fei-Fei et al., 2006; Zhang et al., 2020). In NLP, people find large language models (LLMs) are naturally few-shot learners (Brown et al., 2020), and thus tuning or optimizing prompts becomes extremely important to improve the in-context learning performance of LLMs (Zhou et al., 2022; Sordoni et al., 2023). More recently, within SciML, a different operator learning strategy, termed "in-context operator learning," has been proposed (Yang et al., 2023a;b; Liu et al., 2023). During both training and inference, the neural operator is asked to make predictions by explicitly leveraging so-called "demo" examples (pairs of physical parameters and simulated solutions). This approach provides a balance between model generalization and addressing data scarcity in the scenario of OOD testing.

---

[1]As with others efforts motivated by the goal of foundational models for SciML, the data and scaling used in this paper certainly lack sufficient diversity to categorize the resulting models as "foundational." However, our work marks the first extensive implementation of successful multiple nonlinear physics pretraining for spatiotemporal systems.

## B  PDEs

We consider two time-independent PDEs (Poisson, Helmholtz) and two time-dependent PDEs (Reaction-Diffusion, Navier-Stokes), as depicted below.

1. *Poisson*: We consider a two-dimensional (2D) elliptic PDE that arises in various physical situations, with periodic boundary conditions within a spatial domain $\Omega = [0,1]^2$:

$$-\mathrm{div}\, \boldsymbol{K}\nabla u = f \quad , \tag{1}$$

   where $u(x)$ represents the solution, $f(x)$ acts as the source (e.g., forcing) function, and $x$ denotes spatial coordinate. The diffusion coefficient tensor, denoted by $\boldsymbol{K}$, is employed to measure the physical properties of this system.

2. *Helmholtz*: We consider the 2D inhomogeneous Helmholtz equation, which is a time-independent form of the wave equation, with periodic boundary conditions and a spatial domain $\Omega = [0,1]^2$. The governing equation is given by

$$-\Delta u + \omega u = f \quad \text{in } \Omega, \tag{2}$$

   where $u(x)$ denotes the solution, $f(x)$ represents the source function, and $\omega > 0$ is the wavenumber (that defines the dynamic properties of the Helmholtz equation). This system produces high-frequency large wavenumber oscillatory patterns, which poses challenges in terms of generalization.

3. *Reaction-Diffusion:* The 2D Reaction-Diffusion (RD) equation involves the interaction between two nonlinear coupled variables, i.e., the activator $u(t, x, y)$ and the inhibitor $v(t, x, y)$. The RD equation is given by

$$\begin{aligned} \partial_t u &= D_u(\partial_{xx}u + \partial_{yy}u) + R_u, \\ \partial_t v &= D_v(\partial_{xx}v + \partial_{yy}v) + R_v, \end{aligned} \tag{3}$$

   where $\{D_u, D_v\}$ are diffusion coefficients for $u$ and $v$, respectively. $R_u$ and $R_v$ are reaction functions, which are defined as the Fitzhugh-Nagumo equation (Klaasen & Troy, 1984):

$$\begin{aligned} R_u &= u - u^3 - k - v, \\ R_v &= u - v. \end{aligned} \tag{4}$$

   The spatial domain considered for the 2D RD equation is $\Omega = [-1,1]^2$, and the time duration is $t \in (0, 5]$.

4. *Navier-Stokes Equation:* Lastly, we consider the 2D incompressible Navier-Stokes equation in vorticity form on the unit torus, which is formulated as

$$\begin{aligned} \partial_t w(x,t) + u(x,t) \cdot \nabla w(x,t) &= \nu \Delta w(x,t) + f(x), \\ \nabla \cdot u(x,t) &= 0, \\ w(x,0) &= w_0(x). \end{aligned} \tag{5}$$

   Here, the velocity $u$ and the vorticity $w$ are defined within the time duration $[0, T]$ and the spatial domain $[0,1]^2$ (the vorticity can also be formulated as $w = \nabla \times u$), and $w_0$ is the initial vorticity. Moreover, the coefficient $\nu$ signifies the viscosity of a fluid, and $f(x)$ is the external forcing function. Periodic boundary conditions are employed in this system.

## C  MODEL ARCHITECTURES

We consider two popular architectures for fair comparisons with previous works. These are encoder-decoder architectures designed to reconstruct the original input given partial observations, where the encoder maps observed unlabeled PDE data to a latent space, and the decoder reconstructs the original conditions.

**Fourier Neural Operator.** FNO targets learning PDE data in the Fourier space. The original model backbone (encoder) employs Fourier transform and learns lower Fourier modes with linear transforms. The FNO backbone outputs features back to the spatial domain (i.e., the embeddings are on the pixel level). We refer readers to the original paper for details (Li et al., 2021a;b).

- *Pretraining*: We build the decoder to be identical to the encoder (except for the input/output dimension). Unlabeled PDE data are randomly masked at the pixel level.

- *Fine-tuning*: After the pretraining, we discard the decoder, and we follow the original design to append two fully-connected layers (with ReLU activations) to predict final spatial-wise solutions.

**Transformer.** Transformers, which mainly employ self-attention and linear transform blocks, have been shown to be promising in both NLP and CV (Vaswani et al., 2017; Dosovitskiy et al., 2020). Different from FNO, which directly operates on grids, transformers tokenize and group grids into patches, i.e., each tokenized patch embeds a local neighborhood of subgrids. We follow the 3D transformer architecture of Video-MAE (Tong et al., 2022b; McCabe et al., 2023). Our encoder embeds patches by a linear projection with added positional embeddings (just as in a standard ViT), and then it processes the resulting set via a series of Transformer blocks. For the transformer, the unlabeled PDE data are randomly masked at the patch level.

- *Pretraining*: For the transformer encoder, we only apply it on the subset of tokens that are visible (i.e., unmasked patches), and masked patches are removed. This allows us to train encoders efficiently. The input to the MAE decoder is the full set of tokens consisting of (i) encoded visible patches, and (ii) the mask token. The mask token is a shared and learned vector that indicates the presence of a missing patch to be predicted. We add positional embeddings to all tokens in this full set. Without this, mask tokens would have no information about their location in the input. Following (He et al., 2021; Tong et al., 2022a), we adopt an asymmetric design where the decoder is more lightweight (shallower and narrower) than the encoder.

- *Fine-tuning*: After the pretraining, the decoder is preserved during fine-tuning, since we need to reconstruct the tokenized patches back to the input.

## D  DETAILED EXPERIMENT SETTINGS

### D.1  UNLABELED PDE DATA.

1. For Poisson and Helmholtz (Li et al., 2021a; Subramanian et al., 2023), we consider coefficients, sourcing functions, and coordinates as unlabeled PDE data. These data are controlled by the physical parameters of PDE systems (diffusion coefficients, wavenumber).

2. For the Diffusion-Reaction and Navier Stokes problems, we follow PINO (Li et al., 2021b) and PDEBench (Takamoto et al., 2022) to use snapshots without temporal dynamics as the unlabeled PDE data, controlled by diffusion coefficients and Reynolds number, respectively.

### D.2  DISTRIBUTIONS OF UNLABELED PDE DATA.

In Table 1, we summarize the distribution of our unlabeled PDE data during pretraining, fine-tuning, and OOD inference with in-context learning. Ranges of these physical parameters are inspired by Subramanian et al. (2023). During pretraining, we consider a wide distribution of unlabeled PDE data. When training (fine-tuning) our model, we consider in-distribution unlabeled PDE data. Finally, we test our in-context learning method with OOD settings[2].

For Helmholtz OOD, we choose a narrow range of coefficients ([15, 20]) mainly because its solution is very sensitive to the wavenumber, and FNO's performance significantly drops when we move to more extreme OOD settings.

---

[2]For Navier Stokes from PDEBench, we use the original data. We could not generate our own pretraining/finetuning data with different ranges of physics parameters, due to a possible mismatch of the provided configuration files and the version of Phiflow used in PDEBench (see GitHub issue at https://github.com/pdebench/PDEBench/issues/36). We are still contacting PDEBench and working on it.

Table 1: Ranges of physical parameters (integers) for unsupervised pretraining, training (fine-tuning), and out-of-distribution (OOD, for in-context learning).

| Physics Parameters | Poisson (diffusion) | Helmholtz (wave number) | Naiver Stokes (Reynolds number) |
|---|---|---|---|
| Pretraining | [1, 20] | [1, 20] | {100, 300, 500, 800, 1000} |
| Training (or Fine-tuning) | [5, 15] | [5, 15] | 300 |
| Out-of-Distribution Testing | [15, 50] | [15, 20] | 10000 |

## D.3 HYPERPARAMETERS

We summarize our hyperparameters used during pretraining and fine-tuning/training in Table 2. These hyperparameters strictly follow previous works (Li et al., 2021a; Subramanian et al., 2023; Takamoto et al., 2022; Li et al., 2021b; McCabe et al., 2023).

In our experiments, we trained models three times with different random seeds and provided error bars.

Table 2: Hyperparameters for pretraining and training/fine-tuning. "N.S.": 2D Incompressible Navier-Stokes. "DAdapt": adaptive learning rate by D-adaptation (Defazio & Mischenko, 2023). "ns": total number of simulated training samples.

| Stage ↓ | PDEs → | Poisson | Helmholtz | Reaction-Diffusion | N.S. (PINO) | N.S. (PDEBench) |
|---|---|---|---|---|---|---|
| Pretraining | Number of Samples | 46,080 | 46,080 | 76,760 | 57,545 | 713,286 |
| | Learning Rate | $1 \times 10^{-3}$ | $1 \times 10^{-3}$ | $1 \times 10^{-3}$ | $1 \times 10^{-3}$ | DAdapt |
| | Batch Size | 32 | 32 | 5 | 2 | 8 |
| | Resolution ($H \times W$) | 64×64 | 64×64 | 128×128 | 128×128 | 512×512 |
| | Epochs/Iterations | 500 epochs | 500 epochs | 500 epochs | 100,000 iters | 500 epochs |
| Training/Fine-tuning | Learning Rate | $1 \times 10^{-3}$ | $1 \times 10^{-3}$ | $1 \times 10^{-3}$ | $1 \times 10^{-3}$ | DAdapt |
| | Batch Size | min(32, ns) | min(32, ns) | 5 | 2 | 8 |
| | Resolution ($H \times W$ or $H \times W \times T$) | 64×64 | 64×64 | 128×128×10 | 64×64×1 | 512×512×15 |
| | Epochs/Iterations | 500 epochs | 500 epochs | 500 epochs | 50,000 iters | 500 epochs |
| | Rollouts | N/A | N/A | 91 | N/A | 1 |

## E EXAMPLE OF SIMULATION COSTS

In Table 3, we demonstrate the cheap simulation of only unlabeled PDE data, versus simulating both unlabeled PDE data and solutions, on 2D incompressible Navier-Stoke on PINO Dataset (Li et al., 2021b) and Reaction-Diffusion on PDE-Bench (Takamoto et al., 2022). We can see that unlabeled PDE data are extremely cheap to simulate. Therefore, our pretraining method can boost the performance and meanwhile save the heavy cost of data simulations.

Table 3: Simulation time costs on 2D Incompressible Navier-Stokes ("N.S.") on PINO Dataset and Reaction-Diffusion ("R.D.") on PDE-Bench. "$Re$": Reynolds number. "$D_u, D_v$": diffusion coefficients.

| Data | Physical Parameters | Unlabeled PDE Data (s) | Data+Solutions (s) | Data Size | CPU | GPU |
|---|---|---|---|---|---|---|
| N.S. | $Re = 100$ | 5499.32 | 11013.90 | $20 \times 100 \times 21 \times 512 \times 512$ | 1 AMD EPYC 7763 | 1 NVIDIA A100 (40GB) |
| | $Re = 300$ | 3683.02 | 7625.82 | | | |
| | $Re = 500$ | 4059.71 | 8963.39 | | | |
| | $Re = 800$ | 4829.3 | 10811.15 | | | |
| | $Re = 1000$ | 4957.24 | 10788.69 | | | |
| R.D. | $D_u = 1 \times 10^{-3}, D_v = 5 \times 10^{-3}$ | 29.65 | 6657.34 | $1000 \times 101 \times 128 \times 128 \times 2$ | 1 AMD EPYC 7763 | N/A |

## F    Pseudocode of Test-time In-context Learning

**Algorithm 1:** Pseudocode of Test-time In-context Learning in a PyTorch-like style.

1  **Data resolution:** x-axis ($W$), y-axis ($H$), output temporal steps ($T$), output channel dimensions for the solution ($C_{out}$).
2  **Input:** Query input ($x$). Paired unlabeled PDE data ($X$) and solutions ($Y \in \mathbb{R}^{J \times H \times W \times T \times C_{out}}$) as $J$ demos. Trained Neural Operator Model $\mathcal{M}$. TopK ($k$) demo solutions to aggregate.
3  $\hat{y} = \mathcal{M}(x)$                                                       ▷ Shape: $H \times W \times T \times C_{out}$
4  $\hat{Y} = \mathcal{M}(X)$                                                     ▷ Shape: $J \times H \times W \times T \times C_{out}$
5  $\hat{\delta} = \hat{y}.\text{reshape}(-1, 1, C_{out}) - \hat{Y}.\text{reshape}(1, -1, C_{out})$                    ▷ Shape: $H \times W \times T \times (J \cdot H \cdot W \cdot T) \times C_{out}$
6  $\hat{\delta} = \text{absolute}(\hat{\delta}).\text{sum}(-1)$
7  index = $\text{argsort}(\hat{\delta}, -1)[:, :, :, : k]$         ▷ Spatial and temporal selection of demos similar to the query.
8  $\hat{y}_{icl} = \text{take\_along\_dim}(Y.\text{reshape}(-1, C_{out}), \text{index})$ ▷ Shape: $H \times W \times T \times C_{out} \times k$. Spatial and temporal aggregation of solutions from similar demos.
9  **Return:** $\hat{y}_{icl}.\text{mean}(-1)$

## G    Benefits of Unsupervised Training

Pretraining on unlabeled PDE data is beneficial beyond achieving better performance with less simulated PDE data. *First*, we find that pretraining on unlabeled PDE data can strongly regularize the model against overfitting, without the need for any early-stopping heuristics (we show our smaller generalization gaps in Figure 5 (a-e)). *Second*, unsupervised pretraining can also help models extract useful representations for subsequent operator learning. In Figure 5 (f), we find that with even pre-extracted features (i.e. model's encoder is fixed during fine-tuning), our neural operators can still outperform baselines (trained from scratch).

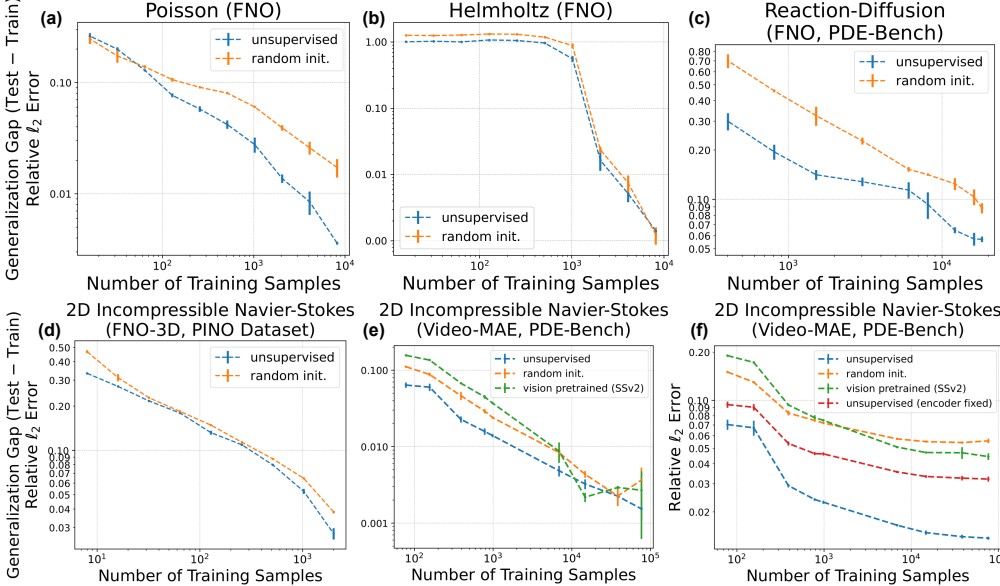

Figure 5: Benefits of unsupervised pretraining: 1) Universally reduced overfitting (i.e. smaller generalization gaps) on all PDEs (**a** to **e**); 2) Meaningful representations: pretraining Video-MAE with fixed encoder (red line) can extract meaningful features from unlabeled PDE data and outperform the baseline and the vision-pretrained model (**f**).

## H    Visualizations of In-context Learning

We also show visualizations of in-context learning in Figure 6. In this visualization, we find that the range of numerical solutions (e.g. values in colorbars) predicted by in-context learning become closer to the target.

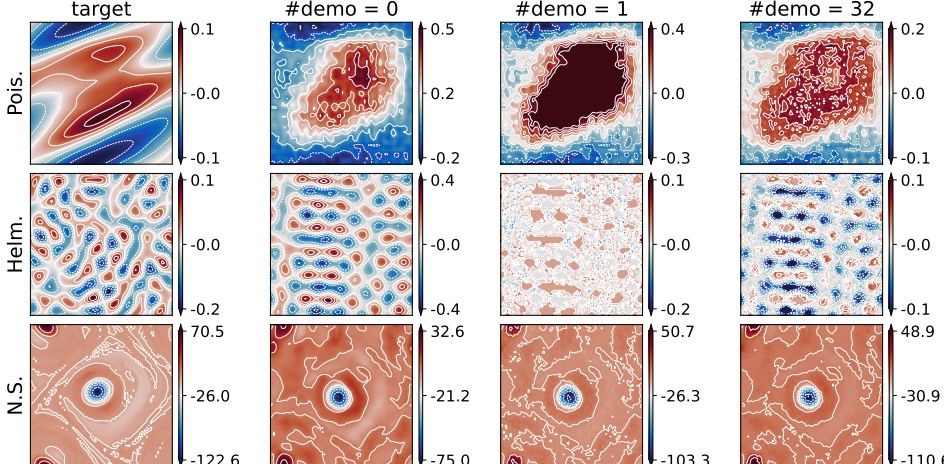

Figure 6: Visualizations of in-context learning for FNO in OOD testing, where ranges of solutions predicted by in-context learning (min/max of each snapshot, reflected in colorbars) become closer to the target.

## I  BENEFITS OF IN-CONTEXT LEARNING

We try to understand the benefit of in-context learning by decomposing the relative MSE error into "scale" and "shape". "Scale" is the slope of a linear regression between targets and predictions (closer to 1 the better), indicating the alignment of the range of model outputs with targets. "Shape" is the normalized relative MSE (i.e. model outputs or targets are normalized by their own largest magnitude before MSE), indicating the alignment of scale-invariant spatial/temporal structures. We show results in Figure 7. We find that the benefit of in-context learning lies in that the scale of the model's output keeps being better calibrated ("scale" being closer to 1) when adding more demos.

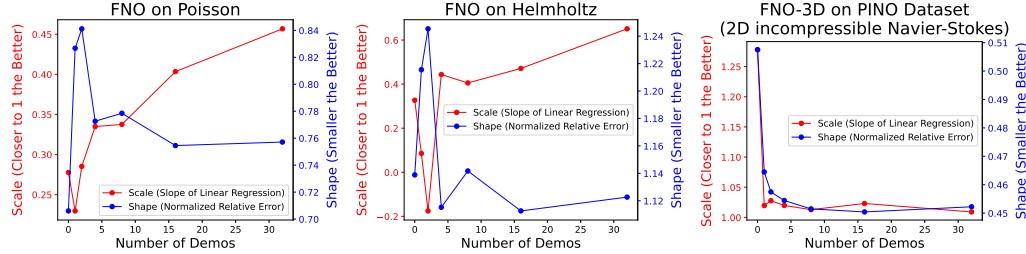

Figure 7: Benefits of in-context learning. To analyze the benefit of in-context learning in complicated PDE systems, we decompose the relative MSE error into "Scale" and "Shape". "Scale" indicates the alignment of the range of model outputs with targets (closer to 1 the better), via the slope of a linear regression. "Shape" indicates the alignment of scale-invariant spatial/temporal structures via normalized relative MSE (i.e. model outputs or targets are normalized by their own largest magnitude before MSE). We find that the benefit of in-context learning lies in that the scale of the model's output keeps being calibrated (red line being closer to 1) when adding more demos.

## J  ABLATION STUDIES

### J.1  MAGNITUDE OF PERTURBATIONS DURING PRETRAINING.

We further study the optimal magnitude of perturbations during pretraining: "Mask Ratio" (1 indicates no visible grids, and 0 means no masks); and "Blur Sigma" for the variance of Gaussian kernel for blur (larger the more degradation). We show our results in Table 4, 5, 6, 7. For example, on 2D incompressible Navier-Stokes with FNO, as shown in Table 7, we can find that when training with a low volume of data, we should use much stronger perturbations (high masking ratios and strong blur), whereas a high volume of data only requires mild perturbations.

Table 4: Best choice of mask ratio and blur sigma for pretraining on Poisson equation.

| #Samples | Mask Ratio | Blur Sigma |
|---|---|---|
| 16 | 0 | 0 |
| 32 | 0 | 0∼1 |
| 64 | 0 | 0∼1 |
| 128 | 0 | 0∼1 |
| 256 | 0 | 0∼1 |
| 512 | 0 | 0∼1 |
| 1024 | 0 | 0∼1 |
| 2048 | 0 | 0∼1 |
| 4096 | 0 | 0∼1 |
| 8192 | 0 | 0∼1 |

Table 5: Best choice of mask ratio and blur sigma for pretraining on Helmholtz equation.

| #Samples | Mask Ratio | Blur Sigma |
|---|---|---|
| 16 | 0.2 | 0∼1 |
| 32 | 0.2 | 0∼1 |
| 64 | 0.2 | 0∼1 |
| 128 | 0.2 | 0∼1 |
| 256 | 0.6 | 0∼0 |
| 512 | 0.6 | 0∼0 |
| 1024 | 0.6 | 0∼0 |
| 2048 | 0 | 0∼1 |
| 4096 | 0 | 0∼1 |
| 8192 | 0 | 0∼0.5 |

Table 6: Best choice of mask ratio and blur sigma for pretraining on 2D Diffusion-Reaction equation.

| #Samples | Mask Ratio | Blur Sigma |
|---|---|---|
| 404 | 0 | 0∼1 |
| 808 | 0 | 0∼1 |
| 1515 | 0 | 0∼1 |
| 3030 | 7 | 0∼2 |
| 6060 | 9 | 0∼1 |
| 8080 | 3 | 0∼1 |
| 12120 | 0 | 0∼1 |
| 16160 | 0 | 0∼1 |
| 18180 | 0 | 0∼1 |

Table 7: Best choice of mask ratio and blur sigma for pretraining on 2D incompressible Navier-Stokes.

| #Samples | Mask Ratio | Blur Sigma |
|---|---|---|
| 8 | 0.7 | 0∼4 |
| 16 | 0.7 | 0∼4 |
| 32 | 0.7 | 0∼4 |
| 64 | 0 | 0∼1 |
| 128 | 0 | 0∼1 |
| 256 | 0.05 | 0 |
| 512 | 0.05 | 0 |
| 1024 | 0.05 | 0 |
| 2000 | 0.05 | 0 |

## J.2 ABLATION OF THE NUMBER OF PRETRAINING SAMPLES.

As shown in Table 8, the more unlabeled PDE data we use for pretraining, the better quality the pretrained model will be.

Table 8: More unlabeled PDE data improve the quality of pretraining. FNO on 2D incompressible Navier-Stokes, pretrained with mask ratio as 0.7.

| #Pretraining Samples | 2000 | 57545 |
|---|---|---|
| Relative $\ell_2$ Error | 0.3594 | 0. 3246 |

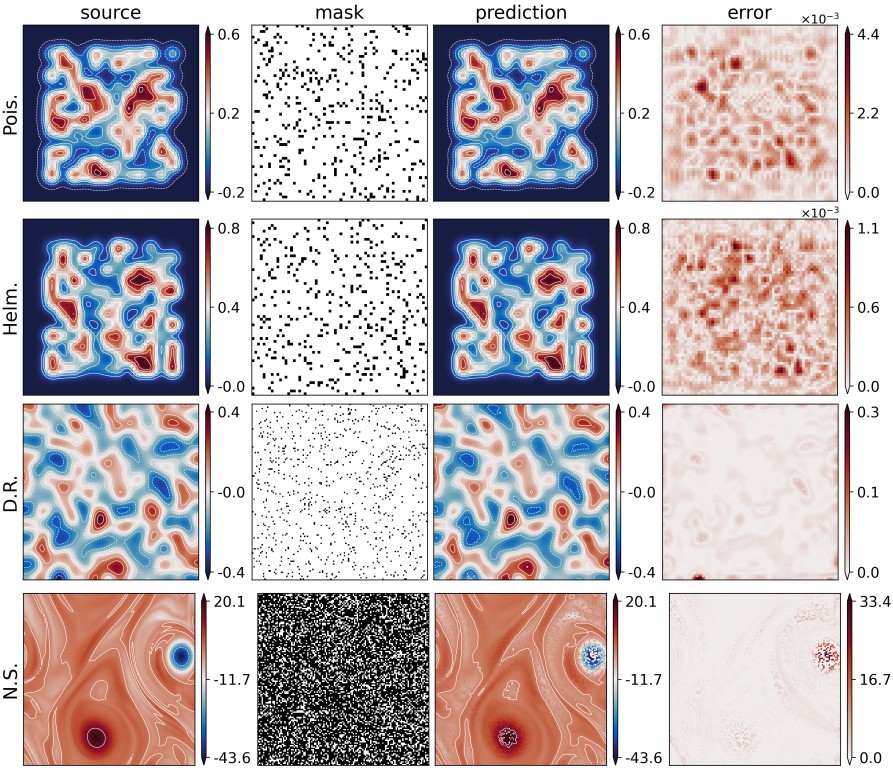

Figure 8: Visualization of reconstructions of unlabeled PDE data on the Poisson ("Pois."), Helmholtz ("Helm."), 2D Diffusion-Reaction ("D.R."), and 2D incompressible Navier-Stokes ("N.S.") equations during MAE pretraining. (Mask ratio: 0.1 for Poisson, Helmholtz, and 2D Diffusion-Reaction equations; 0.7 for incompressible Navier-Stokes.) In masks, only white areas are visible to the model during pretraining.

## K  VISUALIZATION OF MAE PRETRAINING

### K.1  MAE PRETRAINING ON DIFFERENT PDES

To demonstrate the efficacy of our MAE-based pretraining, we show the unlabeled PDE data and its reconstructed version in Figure 8 (MAE pretraining on different PDEs) and Figure 9 (MAE pretraining on 2D incompressible Navier-Stokes with varying mask ratios). We can see that all inputs are accurately reconstructed with low errors and similar patterns.

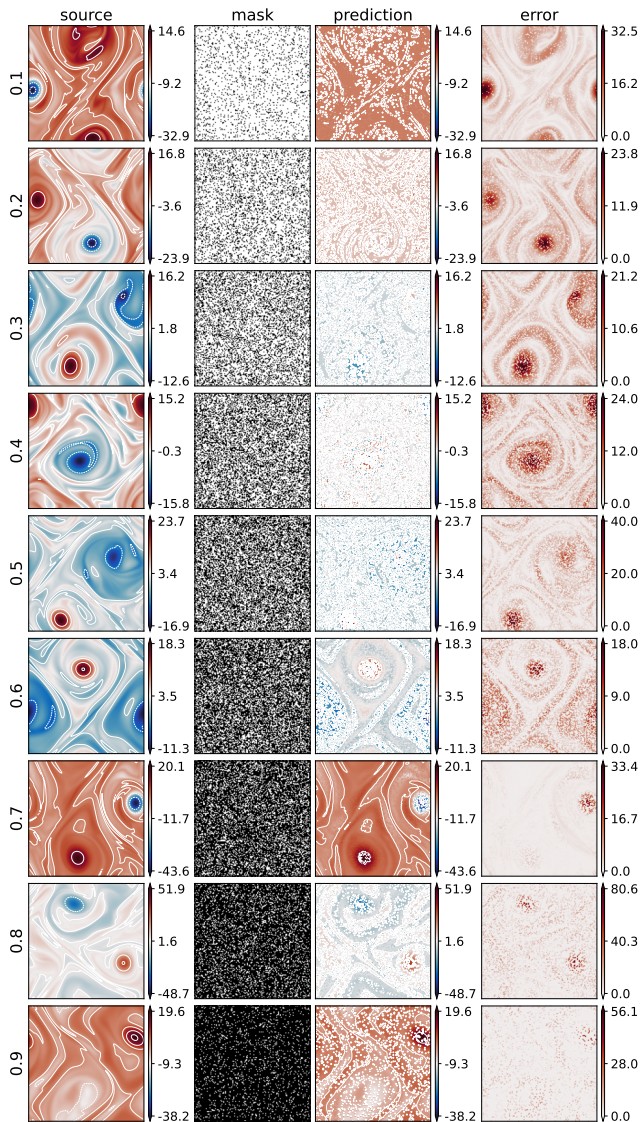

Figure 9: Visualization of reconstructions of unlabeled PDE data on the 2D incompressible Navier-Stokes equations during MAE pertaining with mask ratio from 0.1 to 0.9.

