# OpenReview forum: "Data-Efficient Operator Learning via Unsupervised Pretraining and In-Context Learning"
_ICLR.cc/2024/Workshop/AI4DiffEqtnsInSci — AI4DiffEqtnsInSci @ ICLR 2024 Poster_

### Official Review · Reviewer_93FV · 2024-02-25
**A promising contribution for reaching data efficiency, low computational costs and to avoid overfitting.**

**Rating:** 7
**Confidence:** 2

**Review:**

This work is well written and it is a significant state of the art contribution. It is well supported with up-to-date literature on the topic and the results look very impressive in terms of the magnitude of the errors.
I feel that more details should have been included in the explanation of the methods to be a more appealing work.

Pros:
* The authors clearly highlight the novelty of their contribution and the accuracy of the methods proposed.
* The methods achieve data efficiency and low computational costs.
* It is a nice example of how a model can be trained when having low amounts of data avoiding overfitting.
* The magnitude of the errors achieved after the training is impressive.

Cons:
* I felt the lack of details in the methods. It would have been nice to add more explanations/reasoning about the selection of the input data and the split of the data into training/validation/testing samples.
* It seems that they did not conduct a hyperparameter tuning. I wonder if the results would have changed significantly with different hyperparameters.
* More context about the physical meaning of the parameters in the methods to inform the machine learning-based model is also missing. For instance, the ranges shown in Table 1 Section D2 are not properly explained. I feel more discussion about this topic is needed because they mention in the abstract the promising venues of coupling machine learning methods and physical domain-specific insights.

---

### Official Review · Reviewer_WiKE · 2024-02-28
**Insufficient Depth, as too Many Topics are Combined into one Paper**

**Rating:** 4
**Confidence:** 5

**Review:**

##  Summary of Paper
The paper aims to demonstrate two things: First the authors aim to reduce the simulation-cost in producing vasts amounts of examples to train neural operators. To this end, they propose to create data examples that do not require to solve the differential equation at hand. On this data they train the neural operator to perform common pre-training tasks such as gap-filling, de-noising or de-blurring.
Second, the paper proposes in-context learning for neural operators. The motivation is taken from the emerging abilities of LLM that can solve new tasks when prompted with very few examples. Specifically, they propose to find examples with similar **outputs** and replace the prediction for the sample at hand by the average of the sample and the similar examples.
The authors present empirical evaluations for both of these methods.
Since both the contributions are mostly independent, I will in the following refer to the first as Pre-Training (PT) and the second as In-Context-Learning (ICL).


## Fit for this Workshop
I think that both contributions are interesting for this workshop. While I think that the ICL contribution might be more uncertain overall, I think that the PT contribution will be of great interest to a lot of participants at this workshop.

## Evaluation of the General Structure and Idea
### Pre-Training
I think the pretraining topic is interesting. The authors claim that "Unsupervised pretraining on unlabeled data has never been explored in PDE operator learning in SciML." Independent of, whether this claim is true, I think pretraining for PDE operator learning SciML is not that different to PDE operator learning in other domains and the solutions proposed in this paper are not different to pretraining of neural networks in general. To this end, the article needs a discussion of related  work. I do not agree with the authors that this discussion can be put into the appendix, especially if it is not referenced in the article once.
Further, the explanations in Section 2.1.1 need more explanations. Especially the first bullet point is still unclear. On the other hand, The statement of the second-order linear differential equation does not add much to understanding the paper.
Finally, empirical evaluation is way to short. No details are provided. In the current form, The empirical results are difficult to interpret and almost impossible to reproduce as no information on the architectures or training algorithms is given. It is unclear which pretraining and how mch pretraining is done in each example. This is especially important, as the random initialization results fall short of the values of other publications. For example [1] reports an error of 0.005 at 10 000 examples for the Poisson equation beating the numbers reported for the random initialization by an order of magnitude and being on par with the pretraining.
The description of the Proxy tasks (2.1.2) is nice and I appreciate the fact that the authors do not only introduce the tasks but share their intuition why these tasks should improve the performance.

### In-Context-Learning
I am not sure about this idea of ICL. I understand the motivation, but I see a big discrepancy between the motivation and the actual implementation. In the current form, the method seems to be similar to imposing a prior on the distributions where predictions by the operator are corrected towards areas of high density in the prediction space. If the solution of the PDE is continous, this is further similar to test time augmentation (eg. [2]). I think that both of these links need to be discussed and evaluated.
The description in Section 2.2 is way to short and general. The empirical results have the same shortcomings as the empirical evaluations of the PT approach. They need a lot more explanation and introduction than provided by the authors. In the current form, the results are not convincing.

## Structure and Length of the Paper
I know that it is challenging to explain a complex topic in four pages. This makes it even more difficult to put two paper into four pages. I think the paper is not long enough to discuss both topics well enough to allow the reader to follow and unfortunately, I do not think that any of the two proposed advances are well enough discussed to be helpful to the reader. This is underpined by the fact that the authors provide multiple Appendices (A1, A2, A3, B, C, D1, D2, D3, E, F, G, H, I, J1, J2, K1; 9pages) only three of which are ever referenced in the paper.
I would recommend to focus on one of the two topics instead.

## Conclusion
In conclusion I think the paper tries to do to much on four pages to allow for the needed scientific rigor. I find especially the PT topic interesting but I can not follow the authors arguments and conclusions in this paper. I think that the discussion of the pre training tasks in 2.1.2 is a highlight but especially the empirical evaluation is to poorly described to understand whether there is any effect. Therefore, while I think the underlying research is interesting and might be well done, the paper in the current state is not good enough. I recommend to reject this paper.

##References
[1] Hasani E, Ward RA. Generating synthetic data for neural operators. arXiv preprint arXiv:2401.02398. 2024 Jan 4.

[2] Perez F, Vasconcelos C, Avila S, Valle E. Data augmentation for skin lesion analysis. InOR 2.0 Context-Aware Operating Theaters, Computer Assisted Robotic Endoscopy, Clinical Image-Based Procedures, and Skin Image Analysis: First International Workshop, OR 2.0 2018, 5th International Workshop, CARE 2018, 7th International Workshop, CLIP 2018, Third International Workshop, ISIC 2018, Held in Conjunction with MICCAI 2018, Granada, Spain, September 16 and 20, 2018, Proceedings 5 2018 (pp. 303-311). Springer International Publishing.

---

### Meta-Review · Area_Chair_ADfa · 2024-02-28

**Recommendation:** Accept (Poster)

**Metareview:**

Dear Authors,

Thank you for submitting the draft.

Both reviewers agree that the presented work is interested. However, both reviewers do also raise some major points of concern, especially regarding the clarity of the presentation. It is expected that authors will be addressing comments by the reviewers in the final draft.

regards

AC

---

### Decision · Program_Chairs · 2024-02-29

Accept (Poster)